# Motivation of nursing undergraduate program entrants: A scoping review protocol

**Leonila Santos de Almeida Sasso** [1]*, **Ana Caroline dos Santos Costa**[2], **Ana Maria Rita Pedroso Vilela Torres de Carvalho Engel**[2], **Emília Batista Mourão Tiol**[2], **Fabrício Renato Teixeira Valença**[1], **Natalia Almeida de Arnaldo Silva Rodrigues Castro**[1], **João Daniel de Souza Menezes**[1], **Cíntia Canato Martins**[3], **Carlos Dario da Silva Costa**[1], **Maria Aurélia da Silveira Assoni**[4], **William Donegá Martinez**[1], **Patrícia da Silva Fucuta**[5], **Vânia Maria Sabadoto Brienze**[1], **Alba Regina de Abreu Lima**[1], **Júlio César André**[1]

**1** FAMERP- Faculty of Medicine of São José do Rio Preto, São José do Rio Preto, Brazil, **2** UNIFUNEC—University Center of Santa Fé do Sul, Santa Fé do Sul, Brazil, **3** UNIP- Paulista University, Paulista, Brazil, **4** HA—Cancer Hospital of Barretos, Barretos, Brazil, **5** FACERES—Faceres Medical School, São José do Rio Preto, Brazil

* leosaalm988@gmail.com

## Abstract

Motivation is of great importance in the teaching-learning process, because motivated students seek out opportunities and show interest and enthusiasm in carrying out their tasks. The objective of this review is to identify and present the information available in the literature on the status quo of motivation among nursing program entrants. This is a qualitative scoping review study, a type of literature review designed to map out and find evidence to address a specific research objective, following the Joanna Briggs Institute methodology. The objective was outlined using the PCC (Population, Concept, Context) acronym. The protocol was developed and registered on the Open Science Framework (OSF) platform under DOI 10.17605/OSF.IO/EJNGY. The search strategy and database selection were defined by a library and information science professional together with the authors. The search will be carried out in the following databases: Cumulative Index to Nursing and Allied Health Literature, Literatura Latino Americana e do Caribe em Ciências da Saúde, Lilacs Esp, National Library of Medicine (PubMed), ScienceDirect, Scopus, and the Web of Science platform. The researchers will meet to discuss discrepancies and make decisions using a consensus model, and a third researcher will be tasked with independently resolving any conflicts. Data extraction will involve two independent researchers reviewing each article. Documents such as original articles; theoretical studies; experience reports; clinical study articles; case studies; normative, integrative, and systematic reviews; meta-analyses; meta-syntheses; monographs; theses; and dissertations in English, Portuguese, and Spanish from 2017 to 2023 were included. The results will be presented in tabular and/or diagrammatic format, along with a narrative summary.

**Data Availability Statement:** No datasets were generated or analysed during the current study. All

relevant data from this study will be made available upon study completion.

**Funding:** The author(s) received no specific funding for this work.

## Introduction

In today's health professions, professionals are expected to maintain positive attitudes toward patients. However, there is a noticeable decline in humanistic values among future health professionals, along with reduced idealism and empathy, increased skepticism, and diminished feelings of accomplishment and satisfaction with their career choices. It is crucial for these professionals to adhere to socially accepted standards and professional ethics. They should also support the development of strong professional–patient relationships, thereby enhancing their contribution to the health-disease continuum [1].

Technical mastery is fundamental to saving lives, but it is not enough to maintain quality of life and prevent illness. The challenge, therefore, lies in the balance between technical teaching and humanized practice [2].

The patient of the future is an empowered one, interested in participating in shared decision making. They recognize that even with careful monitoring of their vital signs, they still have family needs. Most likely, they will prefer care from a compassionate healthcare professional and will seek the wise counsel of an experienced advisor who understands them in their natural environment [3].

They will seek someone who respects their dignity and privacy, requiring simple interpretations of isolated data to make sense of their lives. They will look for explanations and meaning, and at times, they will require only a silent witness to their suffering [3–5].

Nursing is a profession. The attributes, behaviors, commitments, values, and objectives that characterize a profession define professionalism, and those who practice the profession are considered professionals [6,7]. Society expects health professionals to be competent, skilled, ethical, humanistic, altruistic, and reliable and that they and their profession promote the health and well-being of individuals and the public [8–10].

Health education has a holistic role in fostering the future professionalism of students, placing them at the core of the process. Students need to be motivated to learn and absorb knowledge. The term "motivation" comes from the Latin movere, meaning to move, signifying "anything that can initiate movement." Based on this understanding, motivation is unique to the individual; shaped by one's experiences, culture, and needs; and is linked to the pursuit of a specific goal [11–13].

According to self-determination theory (SDT), motivation is dynamic and can shift across a continuum, moving from a lack of motivation through externally controlled motivation to self-driven motivation. SDT, when applied to enhancing motivation in the teaching and learning process, is divided into intrinsic motivation (IM), extrinsic motivation (EM), and demotivation (DEM) [14,15]. IM and EM can be further categorized into subtypes, each influencing student behavior and learning outcomes differently [16,17].

Motivation plays a crucial role in the teaching and learning process, as it significantly impacts students' learning and development, affecting how much they invest themselves in the process. A motivated and engaged student actively seeks out opportunities and new knowledge, displaying a keen interest and enthusiasm in completing assigned tasks. Motivation is a key component that allows an individual to use their resources to meet their objectives [18–20].

In the context of undergraduate nursing students, research has highlighted high levels of AM leading to enhanced learning and improved academic performance throughout the program, an aspect desirable for success in health sciences programs [21].

There is a relative lack of literature on the motivation of nursing students for learning, as well as on their academic or teaching performance, and on the teaching of nursing overall. The need for highly motivated nurses, which is a significant concern for any healthcare system,

alongside the need to enhance teaching efficiency and the delivery of quality nursing services, necessitates the education of nursing students with sufficient motivation to accumulate a vast amount of information and skills, as well as the eagerness to learn and continuously update their knowledge as the field evolves. All of this has directed research toward the topic of motivation within the nursing field [19].

International research on motivation for learning among nursing students has intensified in the past decade. The primary focus of international research is on identifying the type of motivation that has the most impact on nursing students, including EM and IM, and the positive and negative associations between types of motivation and student grades and academic outcomes [22]. However, there is still a gap in understanding the specific motivations of nursing program entrants, which could provide valuable insights into student retention, academic performance, and future professional commitment.

## Justification

Understanding the status quo of motivation among nursing program entrants can shed light on students' academic profiles and the degree of resilience and perseverance they will need to successfully complete their studies and advance as professionals in the field. Similarly, based on the type of student motivation, it might be possible to forecast their commitment levels, the risk of program dropout, and their academic progression and whether it falls below expectations or exceeds them.

Scoping review is an excellent method for understanding and mapping studies about the type of motivation influencing students' choices regarding enrolling in a nursing program. It also enables an analysis of how broadly this topic is covered, its level of research and study, and its impact. This facilitates summarizing and sharing existing data on the motivation behind nursing students' pursuit of their undergraduate degree, transparently identifying any gaps in knowledge and allowing for an analysis of students' commitment levels, the risk of dropout, and the influence of motivation on academic progress.

Publishing this study protocol is crucial as it provides transparency in the research process, allows for peer review of the methodology before the study is conducted, and helps prevent duplication of efforts in the scientific community. It also serves as a reference point for future researchers interested in the motivation of nursing program entrants and contributes to the overall body of knowledge in nursing education.

## Materials and methods

### Type of study

This is a qualitative scoping review study, a kind of literature review, carried out following the PRISMA guidelines for scoping reviews [23] and adhering to the JBI Scoping Review Methodology Group's guidance on performing scoping reviews [24,25]. We developed the structure of this scoping review protocol, drawing on the open evidence synthesis protocol framework developed by Ghezzi-Kopel and Porciello [26].

### Protocol registration

This protocol is registered with the Open Science Framework (OSF) and can be found at DOI 10.17605/OSF.IO/EJNGY.

### Research question

What is the status quo of motivation among nursing program entrants?

## Objective

To identify and showcase the available literature on the status quo of motivation among nursing program entrants, and to identify the key characteristics of this motivation that may predict their commitment level, risk of program dropout, and academic progression.

## Inclusion criteria

Full-text articles; studies with both quantitative and qualitative methodologies; primary research; original articles; theoretical explorations; experience reports; clinical study papers; case studies; normative, integrative, and systematic reviews; meta-analyses; meta-syntheses; monographs; theses; and dissertations. Books and book chapters published both in print and online in indexed sources, which address the specified question. Languages: English, Spanish, and Portuguese.

## Exclusion criteria

Studies that are incomplete or duplicate, opinion pieces, consensus statements, retractions, editorials, websites, and media advertisements, along with summaries and proceedings of events. Studies whose primary focus falls outside the scope of the research question, or that fail to conform to the specified definition and criteria. Press documents, book reviews, grey literature, videos, documentaries, films, and publications in languages other than English, Spanish, and Portuguese are also excluded.

## Information sources

**Eligible research databases.**   The relevant multidisciplinary databases were chosen and defined by a library and information science professional in collaboration with the authors to find published studies related to our primary research question. For this study, we will conduct research in the following disciplinary databases: Cumulative Index to Nursing and Allied Health Literature, Latin American and Caribbean Health Sciences Literature, Lilacs Esp, National Library of Medicine (PubMed), ScienceDirect, and on the Web of Science platform. These will be complemented with searches in the multidisciplinary database Scopus (Elsevier).

**Research strategy.**   The research strategy uses the PCC (Population, Concept, and Context) acronym to structure the question formulation.

**Population.**   In this study, the target population consists of nursing program entrants, given the scarcity of studies using this approach. The aim is to gain a better understanding of the profile of nursing students and the motivations that might lead them to complete the program.

**Concept.**   Motivation is a driving force that propels an individual to mobilize and gather resources to move forward despite difficulties and to overcome obstacles in order to achieve their goals and objectives, and to complete their tasks or plans. It is a crucial element for students that keeps them focused on their studies.

**Context.**   Nursing program entrants were chosen as nursing education demands students who are motivated and eager to learn. This is because they have to absorb a substantial amount of dense and highly complex knowledge in a short period, while also developing specific skills.

This study's guiding question incorporated elements of the PCC acronym, which guided the establishment of the inclusion criteria for the review.

**Strategy for searching the databases.**   The strategy for searching the databases was collaboratively determined by the researchers and a library and information science profesional. The search strategy is presented in Table 1.

**Table 1. Search strategy.**

| SEARCH STRATEGY | | | |
|---|---|---|---|
| TOPICS AND STRATEGY | TOPIC AND SYNONYMS IN PORTUGUESE (DeCS) | TOPIC AND SYNONYMS IN ENGLISH (MeSH) | TOPIC AND SYNONYMS IN SPANISH (MeSH) |
| TOPIC 1 | **Estudantes de Enfermagem**" OR "Alunos de Enfermagem" OR "Enfermeiras Estudantes" OR "Enfermeiros Estudantes" OR "Estudante de Enfermagem" | "**Students, Nursing**" OR "Pupil Nurses" OR "Student, Nursing" OR "Nurses, Pupil" "Nurse, Pupil" OR "Pupil Nurse" OR "Nursing Student" OR "Nursing Students" | **"Estudantes de Enfermería"** |
| TOPIC 2 | **Motivação" OR "**Desincentivos" OR "Expectativa" OR "Expectativas" OR "Incentivo" OR "Incentivos" OR "Motivações" OR "Motivo" | **Motivation**" OR "Motivations" OR "Disincentives" OR "Disincentive" OR "Expectations" OR "Expectation" OR "Incentives" OR "Incentive" | "Motivación" |
| TOPIC 3 | **Educação em Enfermagem**" OR ""Curso de Enfermagem" OR "Cursos de Enfermagem" OR "Ensino de Enfermagem" | "Education, Nursing" "Nursing Education" OR "Educations, Nursing" OR "Nursing Educations" | "Educación en Enfermería" |

Source: Authors, 2024.

The search covered the period January 2017 to December 2023.

Table 2 lists the databases to be used in the search and your search strategy.

## Data extraction

The data will be independently extracted by two researchers who have reviewed and tested various initial articles identified during the first phase of the search strategy. The extraction will be carried out using Rayyan® extraction processes. The elements for extraction have been chosen to address the research question and inclusion criteria (including PCC). These elements might be updated or revised during the extraction process; for example, to incorporate trends that were not previously identified. Any such iterative changes or modifications will be

**Table 2. Databases to be used in the search.**

| DATABASES USED IN THE SEARCH | |
|---|---|
| DATABASES | STRATEGY |
| ENGLISH Search strategy: (**Cumulative Index to Nursing and Allied Health Literature (CINAHL); National Library of Medicine (PubMed); Science Direct; SCOPUS e Web of Science)** | "**Students, Nursing**" OR "Pupil Nurses" OR "Student, Nursing" OR "Nurses, Pupil" "Nurse, Pupil" OR "Pupil Nurse" OR "Nursing Student" OR "Nursing Students") AND **Motivation**" OR "Motivations" OR "Disincentives" OR "Disincentive" OR "Expectations" OR "Expectation" OR "Incentives" OR "Incentive") AND ("Education, Nursing" "Nursing Education" OR "Educations, Nursing" OR "Nursing Educations" |
| PORTUGUESE/SPANISH search strategy (**Literatura LatinoAmericana e do Caribe em Ciências da Saúde (LILACS)** | (**"Estudantes de Enfermagem**" OR "Alunos de Enfermagem" OR "Enfermeiras Estudantes" OR "Enfermeiros Estudantes" OR "Estudante de Enfermagem"" AND **Motivação" OR "**Desincentivos" OR "Expectativa" OR "Expectativas" OR "Incentivo" OR "Incentivos" OR "Motivações" OR "Motivo" AND **Educação em Enfermagem" OR "**"Curso de Enfermagem" OR "Cursos de Enfermagem" OR "Ensino de Enfermagem") (**"Estudantes de Enfermería"** AND ""Motivación" AND ""Educación en Enfermería") |

Source: Authors, 2024.

**Table 3. Data extraction tool.**

| Review Title: Motivations Guiding Nursing Program Entrants' Choice for the Nursing Undergraduate Degree Program | |
|---|---|
| **Data Extraction Tool** | |
| **A. Researcher in charge of extraction** | |
| **B. Bibliographic information** | |
| author(s), date, title, journal, volume, issue, pages, year of publication | |
| **C. Country of origin of the study** | |
| **D.** Purpose/objectives of the study | |
| **E.** Method(s) | |
| **F. Type of study** <br> Description of the type of study conducted | |
| **G. Population** | |
| Number of participants | |
| Inclusion criteria | |
| Exclusion criteria | |
| Other characteristics | |
| **H. Location of the study** | |
| Institution (location); Characteristics of the location <br> other | |
| **I. Type of motivation identified in the study** | |
| Specific characteristics of the identified motivation | |
| **H. Results found** | |
| Description of the findings | |
| J. Main conclusions/results of the research | |

Source: Authors, 2024.

thoroughly documented and detailed in the review manuscript. Data extraction for each source will be independently conducted by two researchers, who will finalize their analyses in Rayyan® application (software as a service—SaaS or a web application—that can be accessed at https://www.rayyan.ai/) [27]. Before starting, the researchers will undergo a calibration based on eligibility criteria to ensure uniformity in the inclusion or exclusion of articles for the review. Any discrepancies between reviewers will be resolved by a third reviewer, who will make the final decision. Subsequently, all articles that are excluded will be represented in a flowchart, following the Prisma 2020 flowchart model.

The texts chosen by the two reviewers will have specific data collected through a data extraction process using a tool specifically designed for this purpose (see Table 3).

## Data synthesis and graphs

We will present the data using tables, numbers, and/or graphs to illustrate the data elements we have extracted, as previously described. Accompanying this will be a narrative summary that tackles the research question driving this project. Our analysis will primarily focus on the types of motivation observed among nursing program entrants and will include any key findings, trends, patterns, or notable gaps we have identified in the existing literature concerning social media and teaching and learning. Our reporting of the search strategy outcomes and other aspects will adhere to the PRISMA-ScR guidelines [22] (S1 File).

## Supporting information

**S1 File. PRISMA checklist.**
(DOC)

## Acknowledgments

The authors would like to thank Zelia Cristina Regis, a health research librarian (Regional Faculty of Medicine of São José do Rio Preto-FAMERP), for her thoughtful feedback on a preliminary version of this protocol.

The review will take place between May and August 2024.

## Author Contributions

**Conceptualization:** Leonila Santos de Almeida Sasso, Maria Aurélia da Silveira Assoni.

**Data curation:** Patrícia da Silva Fucuta.

**Formal analysis:** Ana Maria Rita Pedroso Vilela Torres de Carvalho Engel.

**Investigation:** Ana Caroline dos Santos Costa.

**Methodology:** William Donegá Martinez.

**Project administration:** Júlio César André.

**Resources:** João Daniel de Souza Menezes, Cíntia Canato Martins.

**Supervision:** Emília Batista Mourão Tiol, Vânia Maria Sabadoto Brienze, Alba Regina de Abreu Lima.

**Validation:** Natalia Almeida de Arnaldo Silva Rodrigues Castro.

**Visualization:** Fabrício Renato Teixeira Valença.

**Writing – original draft:** Leonila Santos de Almeida Sasso.

**Writing – review & editing:** Carlos Dario da Silva Costa.

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
