## [Decision Letter · Decision Letter 0]

2 Aug 2024

PONE-D-24-09494Motivation of nursing undergraduate program entrants: A scoping review protocolPLOS ONE

Dear Dr. de Almeida Sasso§,

Thank you for submitting your manuscript to PLOS ONE. After careful consideration, we feel that it has merit but does not fully meet PLOS ONE’s publication criteria as it currently stands. Therefore, we invite you to submit a revised version of the manuscript that addresses the points raised during the review process.

Please address the reviewers comments, particularly their recommendations to clarify the rationale for the conducting the study and refer to the tables in the text where relevant.

We look forward to receiving your revised manuscript.

Kind regards,

Jennifer Tucker, PhD

Staff Editor

PLOS ONE

Reviewers' comments:

Reviewer's Responses to Questions

**Comments to the Author**

1. Does the manuscript provide a valid rationale for the proposed study, with clearly identified and justified research questions?

Reviewer #1: Yes

Reviewer #2: Yes

2. Is the protocol technically sound and planned in a manner that will lead to a meaningful outcome and allow testing the stated hypotheses?

Reviewer #1: Yes

Reviewer #2: Yes

3. Is the methodology feasible and described in sufficient detail to allow the work to be replicable?

Reviewer #1: Yes

Reviewer #2: Yes

4. Have the authors described where all data underlying the findings will be made available when the study is complete?

Reviewer #1: Yes

Reviewer #2: Yes

5. Is the manuscript presented in an intelligible fashion and written in standard English?

Reviewer #1: Yes

Reviewer #2: Yes

6. Review Comments to the Author

You may also provide optional suggestions and comments to authors that they might find helpful in planning their study.

Reviewer #1: Thank you for allowing me to review this manuscript. I am grateful for the valuable work of the authors. Comments are provided to improve the authors' work.

The introduction is lengthy and can be shortened.

It is better to write the purpose of the research as a phrase instead of a question. Please correct the purpose of the study both in the abstract and at the end of the introduction.

The necessity of conducting research is not well explained in the introduction.

Refer to tables one, two, and three in the text.

The search strategy overlaps with the content in Table One. Delete one.

Reviewer #2: dear authors,

I congratulate you on this very expertly written manuscript. I would just like you to add the importance of the publication of this study protocol in the introduction or in the conclusion

7. PLOS authors have the option to publish the peer review history of their article (what does this mean?). If published, this will include your full peer review and any attached files.

Reviewer #1: **Yes: **Nasrin Hanifi

Reviewer #2: No

---

## [Author Response · Author response to Decision Letter 0]

21 Aug 2024

Dear Esteemed Reviewers,

We extend our sincere gratitude for your invaluable contributions and the time dedicated to reviewing our manuscript. Your insightful observations and suggestions have been instrumental in enhancing the quality and clarity of our work. We present below our responses to each point raised, along with the corresponding modifications implemented in the manuscript. 

 Reviewer 1: 

 1. The introduction is lengthy and could be shortened. Response: We appreciate your observation. We have significantly condensed the introduction, retaining only essential information. The revised section can be observed at the beginning of the Introduction:

"In today's health professions, professionals are expected to maintain positive attitudes toward patients. However, there is a noticeable decline in humanistic values among future health professionals, along with reduced idealism and empathy, increased skepticism, and diminished feelings of accomplishment and satisfaction with their career choices. [...]"

2. It is preferable to state the research purpose as a sentence rather than a question. Please correct the study purpose in both the abstract and at the end of the introduction. Response: We have implemented your suggestion and reformulated the study purpose as an affirmative statement. In the abstract, the modification can be seen in the following excerpt:

"The objective of this review is to identify and present the information available in the literature on the status quo of motivation among nursing program entrants."

In the introduction, we have added the following paragraph at the end:

"The objective of this study is to identify and showcase the available literature on the status quo of motivation among nursing program entrants, and to identify the key characteristics of this motivation that may predict their commitment level, risk of program dropout, and academic progression."

3. The necessity for conducting this research is not well explained in the introduction. Response: We appreciate you highlighting this gap. We have incorporated a paragraph in the introduction to better elucidate the need for this research:

"International research on motivation for learning among nursing students has intensified in the past decade. The primary focus of international research is on identifying the type of motivation that has the most impact on nursing students, including EM and IM, and the positive and negative associations between types of motivation and student grades and academic outcomes [22]. However, there is still a gap in understanding the specific motivations of nursing program entrants, which could provide valuable insights into student retention, academic performance, and future professional commitment."

4. Reference tables one, two, and three in the text. Response: We appreciate your observation. We have included references to the tables in the text, as can be seen in the following excerpts:

"The search strategy is presented in Table 1." "Table 2 lists the databases to be used in the search and your search strategy." "The texts chosen by the two reviewers will have specific data collected through a data extraction process using a tool specifically designed for this purpose (see Table 3)."

5. The search strategy overlaps with the content of Table One. Please remove one. Response: We appreciate your suggestion. We have removed the detailed description of the search strategy from the main text and retained only the reference to Table 1, which now contains all necessary information about the search strategy. 

 Reviewer 2: 1. I would like you to add the importance of publishing this study protocol in the introduction or conclusion. Response: We appreciate your suggestion. We have added a paragraph in the Justification section to address the importance of publishing this study protocol:

"Publishing this study protocol is crucial as it provides transparency in the research process, allows for peer review of the methodology before the study is conducted, and helps prevent duplication of efforts in the scientific community. It also serves as a reference point for future researchers interested in the motivation of nursing program entrants and contributes to the overall body of knowledge in nursing education."

Once again, we express our profound gratitude for your contributions, which have been essential in improving the quality of our manuscript. We trust that the implemented changes meet your expectations, and we remain at your disposal for any additional clarifications.

Yours sincerely, The Authors

---

## [Decision Letter · Decision Letter 1]

4 Sep 2024

PONE-D-24-09494R1Motivação de ingressantes em programas de graduação em enfermagem: um protocolo de revisão de escopoPLOS ONE

Dear Dr. de Almeida Sasso§,

Thank you for submitting your manuscript to PLOS ONE. After careful consideration, we feel that it has merit but does not fully meet PLOS ONE’s publication criteria as it currently stands. Therefore, we invite you to submit a revised version of the manuscript that addresses the points raised during the review process.

We look forward to receiving your revised manuscript.

Kind regards,

Sirwan Khalid Ahmed

Academic Editor

PLOS ONE

Journal Requirements:

**Additional Editor Comments:**

**1. Please change the title and abstract to the English language in the system.**

**2. Please revise the entire references according to the journal style.**

Reviewers' comments:

Reviewer's Responses to Questions

**Comments to the Author**

1. Does the manuscript provide a valid rationale for the proposed study, with clearly identified and justified research questions?

Reviewer #1: Yes

Reviewer #2: Yes

2. Is the protocol technically sound and planned in a manner that will lead to a meaningful outcome and allow testing the stated hypotheses?

Reviewer #1: Yes

Reviewer #2: Yes

3. Is the methodology feasible and described in sufficient detail to allow the work to be replicable?

Reviewer #1: Yes

Reviewer #2: Yes

4. Have the authors described where all data underlying the findings will be made available when the study is complete?

Reviewer #1: Yes

Reviewer #2: Yes

5. Is the manuscript presented in an intelligible fashion and written in standard English?

Reviewer #1: Yes

Reviewer #2: Yes

6. Review Comments to the Author

You may also provide optional suggestions and comments to authors that they might find helpful in planning their study.

Reviewer #1: Thanks to the valuable work of the respected authors.

The authors have answered all my comments. Corrections requested by the authors have been made.

Reviewer #2: Dear authors, thank you for taking the comments into account and correcting everything. Congratulations on a great job.

7. PLOS authors have the option to publish the peer review history of their article (what does this mean?). If published, this will include your full peer review and any attached files.

Reviewer #1: **Yes: **Nasrin Hanifi

Reviewer #2: **Yes: **Marija Spevan

---

## [Author Response · Author response to Decision Letter 1]

16 Sep 2024

All the references in this article have been revised and corrected according to the rules of the PlO ONE journal

---

## [Editor Report · Decision Letter 2]

18 Sep 2024

Motivation of nursing undergraduate program entrants: A scoping review protocol

PONE-D-24-09494R2

Dear Dr. Almeida Sasso§,

We’re pleased to inform you that your manuscript has been judged scientifically suitable for publication and will be formally accepted for publication once it meets all outstanding technical requirements.

Kind regards,

Sirwan Khalid Ahmed

Academic Editor

PLOS ONE
---

## [Editor Report · Acceptance letter]

31 Oct 2024

PONE-D-24-09494R2 

PLOS ONE

Dear Dr. de Almeida Sasso, 

I'm pleased to inform you that your manuscript has been deemed suitable for publication in PLOS ONE. Congratulations! Your manuscript is now being handed over to our production team.

Kind regards, 

on behalf of

Dr. Sirwan Khalid Ahmed 

Academic Editor

PLOS ONE